# RECURSIVE REASONING FOR SAMPLE-EFFICIENT MULTI-AGENT REINFORCEMENT LEARNING

## ABSTRACT

Policy gradient algorithms for deep multi-agent reinforcement learning (MARL) typically employ an update that responds to the current strategies of other agents. While being straightforward, this approach does not account for the updates of other agents within the same update step, resulting in miscoordination and reduced sample efficiency. In this paper, we introduce methods that recursively refine the policy gradient by updating each agent against the updated policies of other agents within the same update step, speeding up the discovery of effective coordinated policies. We provide principled implementations of recursive reasoning in MARL by applying it to competitive multi-agent algorithms in both on and off-policy regimes. Empirically, we demonstrate superior performance and sample efficiency over existing deep MARL algorithms in StarCraft II and multi-agent MuJoCo. We theoretically prove that higher recursive reasoning in gradient-based methods with finite iterates achieves monotonic convergence to a local Nash equilibrium under certain conditions.

## 1 INTRODUCTION

Deep multi-agent reinforcement learning (MARL) research has made strides towards solving practical problems such as cooperative robotics (Ismail et al., 2018), transportation management (Haydari & Yılmaz, 2020) and network traffic optimization (Pi et al., 2024). While deep RL algorithms have garnered impressive results in complex single-agent control problems (Mnih et al., 2015; Tang et al., 2024), multi-agent systems present unique challenges. Roadblocks in MARL research include exploding joint state-action spaces and non-stationarity due to concurrent learning (Li et al., 2009; Barfuss & Mann, 2021). Another significant challenge caused by simultaneous learning in MARL is that each agent's update does not account for the updates of other agents in the same update step, resulting in reduced sample efficiency (Zhang et al., 2021).

In this paper, we propose applying recursive reasoning to refine the policy gradient update, allowing each agent to reason about the change in behavior of other agents. Idealistic multi-agent frameworks assume mutual consistency: the assumption that each agent's beliefs about other agents' behavior and updates are accurate (Robertson, 1936). Due to limited computation, practical multi-agent systems update each agent as if the policy of every other agent is fixed - an assumption known as fictitious play (Foster & Young, 1998). The notion of mutual consistency in the deep MARL setting can be achieved by using the updated policies of other agents in order to recursively refine the policy gradient.

Research in both on-policy and off-policy MARL algorithms have produced impressive results from the extension of single-agent RL methods to incorporate joint state and action information available in the multi-agent setting. In this work, we demonstrate the effectiveness of recursive reasoning in on and off-policy settings in three challenging multi-agent coordination environments. We first introduce a recur-

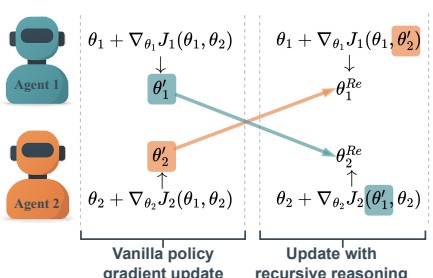

Figure 1: Illustration of recursive reasoning with policy gradients for two agents.

sive on-policy algorithm we term **ReMAPPO** (**Re**cursive **M**ulti-**A**gent **P**roximal **P**olicy **O**ptimization) (based on MAPPO (Yu et al., 2022)) which extends the performance difference lemma (Kakade & Langford, 2002) to model policy distribution updates. We also apply recursive reasoning to off-policy algorithms which utilize the deterministic policy gradient theorem (Silver et al., 2014); we term these implementations **ReFACMAC** (**Re**cursive **FAC**tored **M**ulti-**A**gent **C**entralized policy gradients) and **ReMADDPG** (**Re**cursive **M**ulti-**A**gent **D**eep **D**eterministic **P**olicy **G**radients), based on FAC-MAC (Peng et al., 2021) and MADDPG (Lowe et al., 2017) respectively. Finally, we motivate our method further by conducting a theoretical study of higher recursive reasoning with policy gradients and show that it results in bounded convergence to an $\epsilon$-Nash equilibrium under certain conditions with finite iterates.

## 2 RELATED WORKS

**Policy gradient methods in multi-agent reinforcement learning**   Policy gradient methods often estimate action-values of joint actions taken during training. MADDPG utilizes a multi-agent extension of the deterministic policy gradient (Silver et al., 2014) and exhibits more robust behavior than independent DDPG (Lillicrap et al., 2015). Foerster et al. (2018) propose COMA, which uses a baseline term to reduce centralized gradient noise, improving credit assignment. Similarly, Du et al. (2019) implement a framework that learns a proxy reward in order to discriminatingly credit agents in multi-agent actor-critic methods. Yu et al. (2022) conduct a comprehensive study on MAPPO, a variant of Proximal Policy Optimization (PPO) (Schulman et al., 2017) which conditions its advantage estimation on global state information. While the policy gradient methods mentioned display impressive results, they suffer from instability caused by a lack of mutual consistency, since they don't account for the updates of other agents. To counter this, LOLA (Foerster et al., 2017) utilizes higher-order gradient terms to mutually shape the learning updates of agents in two-player reciprocity-based games. POLA (Zhao et al., 2022) builds on LOLA, reinterpreting it as a proximal operator by penalizing divergence over policy behavior, mitigating LOLA's sensitivity to parameterizations. POLA's viability in complex multi-agent cooperative environments has not yet been sufficiently examined; we show in this work that our implementations of recursive reasoning achieve greater performance and sample efficiency. Unlike POLA, our implementations focus on recursive reasoning via informative policy losses as opposed to computationally expensive higher-order gradient terms or rollouts. M-FOS (Khan et al., 2023) also uses opponent shaping with a recursive paradigm; we do not benchmark it as its meta-game formulation is out of the scope of this paper.

**Recursive reasoning in reinforcement learning**   Recursive reasoning has proven useful in several multi-agent opponent modeling scenarios. Notably, the cognitive hierarchies framework (Camerer et al., 2004) has been combined extensively with deep RL techniques for the training of self-driving vehicles which must cooperate within a heterogeneous population of peer vehicles (Wang et al., 2022; Karimi et al., 2023; Dai et al., 2023). The notion of recursive reasoning in MARL is linked to intuition behind successful human collaboration: social research suggests that humans collaboratively solve complex problems better when they model the decision-making process of other humans (Gurney et al., 2021; Schaafsma et al., 2015).

## 3 PRELIMINARIES

We consider a multi-agent extension of a Markov decision process (Puterman, 2014) known as a Markov game (Littman, 1994), defined by a tuple $\mathcal{G} = \langle \mathcal{I}, \mathcal{S}, \mathcal{A}, \mathcal{P}, \mathcal{R}, \gamma, N, \iota \rangle$. $\mathcal{I} \equiv \{1, \ldots, N\}$ is the set of agents, $\mathcal{S}$ is the state space, $\mathcal{A} \equiv \times_{i \in \mathcal{I}} A_i$ is the joint action space of the agents. At each timestep $t$, agent $i$ (the 'self' agent) samples an action $a_i \in A_i$ from policy $\pi_i(a_i|s)$ parameterized by $\boldsymbol{\theta}_i \in \mathbb{R}^{d_i}$, where $d_i$ is the dimensionality of the parameterization. At each timestep, the 'non-self' agents sample a joint action $\boldsymbol{a}_{-i} \in \times_{j \in \mathcal{I} \setminus i} A_j$ from joint policy $\boldsymbol{\pi}_{-i}(.|s)$ parameterized by joint non-self parameters $\boldsymbol{\theta}_{-i} \in \mathbb{R}^{\sum_{j \in \mathcal{I} \setminus i} d_j}$. The joint action of all agents $\boldsymbol{a}$ from the joint policy $\boldsymbol{\pi}(.|s)$ determines the next state according to the joint state transition function $\mathcal{P}(s'|s, \boldsymbol{a})$. In the case of deterministic policies, we denote the self, non-self, and joint policies as $\mu_i$, $\boldsymbol{\mu}_{-i}$, and $\boldsymbol{\mu}$ respectively. $\mathcal{R} \equiv \{R_1, \ldots, R_N\}$ are the set of agent reward functions. Each agent $i$ has a learning rate $\eta_i$ and receives a reward $r_{i,t}$ at time $t$ according to its reward function $R_i(s, \boldsymbol{a}, s')$. $\gamma$ is the discount factor and $\iota$ is the initial state distribution. We define the joint value function for agent

$i$ as $V_i^{\boldsymbol{\pi}}(s) = \mathbb{E}_{\boldsymbol{\pi},P}\left[\sum_{k=0}^{\infty}\gamma^k r_{i,k}|s\right]$ as the sum of discounted rewards for agent $i$ following the joint policy $\boldsymbol{\pi}$ from state $s$. Similarly, we define the joint action-value function for agent $i$ as $Q_i^{\boldsymbol{\pi}}(s,\boldsymbol{a}) = \mathbb{E}_{\boldsymbol{\pi},P}\left[\sum_{k=0}^{\infty}\gamma^k r_{i,k}|s,\boldsymbol{a}\right]$ as the sum of discounted rewards for agent $i$ after taking joint action $\boldsymbol{a}$ in state s and following the joint policy $\boldsymbol{\pi}$ thereafter. We define the advantage function for agent $i$ as $A_i^{\boldsymbol{\pi}}(s,\boldsymbol{a}) = Q_i^{\boldsymbol{\pi}}(s,\boldsymbol{a}) - V_i^{\boldsymbol{\pi}}(s)$. Each agent aims to maximize its own multi-agent objective, $J_i\left(\boldsymbol{\theta}_i,\boldsymbol{\theta}_{-i}\right) = \mathbb{E}_{s\sim\iota}\left[V_i^{\boldsymbol{\pi}}(s)\right]$. Note that we condition objectives and loss functions on parameters $\boldsymbol{\theta}$ to emphasize that gradient updates are made in weight space, while value functions are denoted with policies $\boldsymbol{\pi}$ to emphasize that actions depend on both weights and the form of the policy distribution.

## 4 RECURSIVE REASONING IN MULTI-AGENT POLICY GRADIENT ALGORITHMS

Multi-agent policy gradient algorithms are primarily concerned with estimating the gradient of the objective in Section 3. Typical estimations of $\nabla_{\boldsymbol{\theta}_i}J_i\left(\boldsymbol{\theta}_i,\boldsymbol{\theta}_{-i}\right)$ respond to the action distribution of other agents before they have made an update, under the assumption of fictitious play. We use $\boldsymbol{\theta}_i'$ to denote agent $i$'s parameters after a naive update:

$$\boldsymbol{\theta}_i' \leftarrow \boldsymbol{\theta}_i + \eta_i\nabla_{\boldsymbol{\theta}_i}J_i\left(\boldsymbol{\theta}_i,\boldsymbol{\theta}_{-i}\right),\forall i \in \mathcal{I}. \tag{1}$$

Once $\boldsymbol{\theta}_i'$ has been obtained for all agents, the update step can be taken once again from the *initial* parameters of each agent while considering the updated policies of the other agents. We use $\boldsymbol{\theta}_i^{Re}$ to denote agent $i$'s parameters after this recursive procedure:

$$\boldsymbol{\theta}_i^{Re} \leftarrow \boldsymbol{\theta}_i + \eta_i\nabla_{\boldsymbol{\theta}_i},J_i\left(\boldsymbol{\theta}_i,\boldsymbol{\theta}_{-i}'\right),\forall i \in \mathcal{I}. \tag{2}$$

Successfully estimating Equation 2 in deep MARL settings is the primary aim of this work. Note that the recursive update of each agent is still only *one* gradient step away from the initial parameters; **the recursive updates are not moving further in weight space, but rather finding a gradient direction that is refined by the updated policies of the other agents.**

### 4.1 REMAPPO

Recall the surrogate loss of Multi-Agent Proximal Policy Optimization (MAPPO):

$$\mathcal{L}_i^{\text{MAPPO}}(\boldsymbol{\theta}_i) = \mathbb{E}_{(s,\boldsymbol{a})\sim\boldsymbol{\pi}}\left[\min\left(r_i(s,a_i),\text{clip}(r_i(s,a_i),1-\epsilon,1+\epsilon)\right)\cdot A_i^{\boldsymbol{\pi}}(s,\boldsymbol{a})\right], \tag{3}$$

where $r_i(s,a_i) = \frac{\pi_i'(a_i|s)}{\pi_i(a_i|s)}$ is the sampling ratio of the updated policy for actions taken in the environment, $\epsilon$ is a clipping boundary, and $A_i^{\pi_i}(s,\boldsymbol{a})$ is the state-conditioned advantage function for policy $\pi_i$. By extending the performance difference lemma (PDL) over updated actions of non-self agents, we derive a new surrogate loss for the algorithm we call ReMAPPO:

$$\mathcal{L}_i^{\text{ReMAPPO}}(\boldsymbol{\theta}_i,\boldsymbol{\theta}_{-i}') = \mathbb{E}_{(s,\boldsymbol{a})\sim\pi_i,\boldsymbol{\pi}_{-i}'}\left[\min\left(r_i'(s,\boldsymbol{a}),\text{clip}(r_i'(s,\boldsymbol{a}),1-\epsilon,1+\epsilon)\right)\cdot A_i^{\boldsymbol{\pi}}(s,\boldsymbol{a})\right], \tag{4}$$

where $\boldsymbol{\theta}_{-i}'$ and $\boldsymbol{\pi}_{-i}'$ are the updated parameters and policies of non-self agents respectively. Note that in both surrogate losses, $A_i^{\boldsymbol{\pi}}(s,\boldsymbol{a})$ is typically estimated using (solely state-dependent) generalized advantage estimation (Schulman et al., 2015). Crucially, we define a new importance sampling ratio $r_i'(s,\boldsymbol{a}) = \frac{\pi_i'(a_i|s)}{\pi_i(a_i|s)}\cdot\frac{\boldsymbol{\pi}_{-i}'(\boldsymbol{a}_{-i}|s)}{\boldsymbol{\pi}_{-i}(\boldsymbol{a}_{-i}|s)}$, which modifies $r_i(s,a_i)$ with the updated joint probability of the non-self policies. Intuitively, this additional joint probability ratio can be seen as a weighting of the original surrogate loss in Equation 3 which places more or less weight on each sample depending on the updates of other agents. If the advantage is positive and other other agents increase their action probabilities, the self-agent is incentivized to increase its probability even more in response. If the advantage is positive but the other agents decrease their respective action probabilities, the degree to which the self-agent will increase its action probability is reduced accordingly. The effect is similarly intuitive in the negative advantage case. A comprehensive derivation of the ReMAPPO surrogate objective is available in Appendix C.

## 4.2 REFACMAC AND REMADDPG

Off-policy gradient algorithms in MARL typically estimate the Deterministic Policy Gradient (DPG) over joint actions. The actor losses for MADDPG and FACMAC take the following respective forms:

$$\mathcal{L}_i^{\text{MADDPG}}(\boldsymbol{\theta}_i, \boldsymbol{\theta}_{-i}) = \mathbb{E}_{s \sim d^{\boldsymbol{\beta}}} \left[ Q_i^{\mu_i}(s, \boldsymbol{\mu}(s)) \right], \tag{5}$$

$$\mathcal{L}_i^{\text{FACMAC}}(\boldsymbol{\theta}_i, \boldsymbol{\theta}_{-i}) = \mathbb{E}_{s \sim d^{\boldsymbol{\beta}}} \left[ \mathcal{F} \left( Q_1^{\mu_1}(s, \boldsymbol{\mu}(s)), ..., Q_i^{\mu_i}(s, \boldsymbol{\mu}(s)), ..., Q_N^{\mu_N}(s, \boldsymbol{\mu}(s)) \right) \right], \tag{6}$$

where $\beta$ is an arbitrary behavior policy, $Q_i^{\mu_i}(s, \boldsymbol{\mu}(s))$ is the action-value function for agent $i$ over joint actions, and $\mathcal{F}$ is a learned mixing function which factorizes a joint action-value function. By incorporating the updates of non-self agents, we reformulate these losses to define the recursive algorithms ReMADDPG and ReFACMAC:

$$\mathcal{L}_i^{\text{ReMADDPG}}(\boldsymbol{\theta}_i, \boldsymbol{\theta}'_{-i}) = \mathbb{E}_{s \sim d^{\boldsymbol{\beta}}} \left[ Q_i^{\mu_i}(s, \mu_i(s), \boldsymbol{\mu}'_{-i}(s)) \right], \tag{7}$$

$$\mathcal{L}_i^{\text{ReFACMAC}}(\boldsymbol{\theta}_i, \boldsymbol{\theta}'_{-i}) = \mathbb{E}_{s \sim d^{\boldsymbol{\beta}}} \left[ \mathcal{F} \left( ..., Q_i^{\mu_i}(s, \mu_i(s), \boldsymbol{\mu}'_{-i}(s)), ..., \right) \right]. \tag{8}$$

Note that while correctly estimating the DPG requires an unbiased action-value estimate, off-policy algorithms such as DDPG Lillicrap et al. (2015) introduce bias in practice via stabilization tricks such as bootstrapping with target networks. Centralized multi-agent algorithms introduce further bias by estimating joint action value functions with off-policy target bootstrapping of non-self agents (Liu et al., 2022; Lowe et al., 2017). In other words, the centralized and factored critics used in MADDPG and FACMAC are trained on actions taken by the behavior policies of the agents, but are used to estimate the joint action-value function given arbitrary actions from the non-self agents during the actor update. By using recursive non-self actions for each agent's joint action-value function, we also estimate the value of joint actions that are arbitrarily distinct from those present during data collection (with the added benefit of mutual consistency with the updates of other agents). Thus, we maintain that ReMADDPG and ReFACMAC introduce no additional bias over their non-recursive counterparts.

# 5 EXPERIMENTS

## 5.1 EXPERIMENTAL SETUP

In this section, we demonstrate the effectiveness of recursive reasoning algorithms across three challenging benchmarks: StarCraft II in JaxMARL (SMAX) (Rutherford et al., 2023), the StarCraft II Multi-Agent Challenge (SMAC) (Samvelyan et al., 2019), and Multi-Agent Multi-Joint dynamics with Contact (MAMuJoCo) (Peng et al., 2021). Note that SMAX and SMAC have different dynamics and are considered separate benchmarks. We benchmark 11 SMAX maps (8 SMACv1-based and 3 SMACv2-based), 8 SMAC maps, and 4 MAMuJoCo maps. These environments present a diverse set of cooperative tasks on which we can compare the sample efficiency and performance of our methods against competitive MARL algorithms.

On SMAX, we compare ReMAPPO against Independent Proximal Policy Optimization (IPPO) (Schulman et al., 2017) and Multi-Agent Proximal Policy Optimization (MAPPO) as on-policy gradient-based methods. We also benchmark Independent Q-Learning (Watkins & Dayan, 1992) (IQL), Value Decomposition Networks (VDN) (Sunehag et al., 2017), and QMIX (Rashid et al., 2020b). Finally, we benchmark POLA (Zhao et al., 2022) as it is the most well-known MARL algorithm which incorporates higher-order opponent updates (in particular, we implement Outer POLA with advantage estimation). We do not benchmark FACMAC and MADDPG in SMAX due to a lack of existing well-tuned implementations in JAX-based environments. A lack of satisfactory results across hyperparameter sweeps (both nominal and those of similar algorithms tuned on SMAX) leads us to believe a comparison in CPU-based environments is more fair.

In SMAC and MAMuJoCo, we compare ReFACMAC against FACMAC, QMIX/COMIX, MAPPO, MADDPG, and POLA. We choose FACMAC as the algorithm upon which to demonstrate the benefits of recursion as FACMAC achieves SOTA performance in these environments. We choose not to benchmark ReMAPPO in these environments, as the low parallelizability and low-data regime of CPU-based environments is not as conducive to on-policy algorithms (as seen by MAPPO's

performance in these benchmarks). We therefore consider it more fair to benchmark ReMAPPO in SMAX as a fair comparison of sample efficiency. Results for ReMADDPG are present in Section 6.3. Note that our results may appear different to previous works (Rashid et al., 2020b;a) since they report median instead of mean win rates, which reduces the impact of failing seeds, especially on difficult maps like Corridor.

## 5.2 EXPERIMENTAL RESULTS

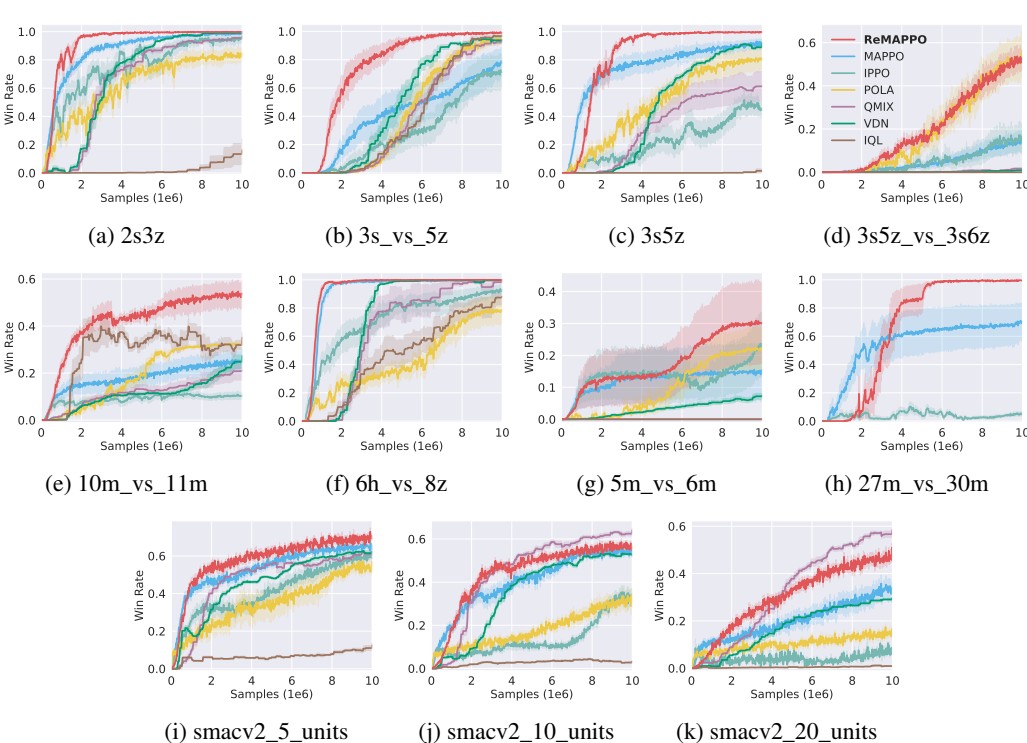

(a) 2s3z  (b) 3s_vs_5z  (c) 3s5z  (d) 3s5z_vs_3s6z

(e) 10m_vs_11m  (f) 6h_vs_8z  (g) 5m_vs_6m  (h) 27m_vs_30m

(i) smacv2_5_units  (j) smacv2_10_units  (k) smacv2_20_units

Figure 2: Mean win rate and standard error of ReMAPPO and baselines on SMAX maps across 10 seeds. Note that 27m_vs_30m is very large and could only be benchmarked with PPO-based methods due to computational constraints.

**ReMAPPO outperforms baselines in SMAX**  Figure 2 compares the test win rate of ReMAPPO against baselines on SMAX. Notably, ReMAPPO performs equal or better than other baselines on 9 out of 11 maps, and is the only algorithm to achieve a 100% win rate in 3s5z and 27m_vs_30m. ReMAPPO also maintains a consistent advantage over MAPPO, demonstrating the viability of recursive reasoning. The effects of recursive reasoning are seen in higher overall performance as well as faster convergence, as ReMAPPO often reaches its maximum performance with fewer samples than other methods. Despite utilizing opponent shaping, POLA fails to match the performance of ReMAPPO in every map except 3s5z_vs_3s6z. This suggests that POLA's formulation for 2-player reciprocity-based games likely does not generalize well to more complex environments with many agents.

**ReFACMAC outperforms baselines in SMAC and MAMuJoCo**  Figure 3 compares the test win rate of ReFACMAC and related baselines in SMAC with a focus on Hard and Super Hard maps. ReFACMAC achieves higher or equal final success rates compared to baselines and typically converges to its maximum win rate with fewer samples. ReFACMAC particularly stands out in Corridor and 3s5z_vs_3s6z, two notoriously difficult maps in which it is the only algorithm to surpass a 50% success rate. Note that MAPPO performs much worse on SMAC maps due to SMAC being CPU-based and less parallelizable, resulting in far less sample availability (it was only possible to obtain 2e6 samples for each map rather than 1e7 in SMAX).

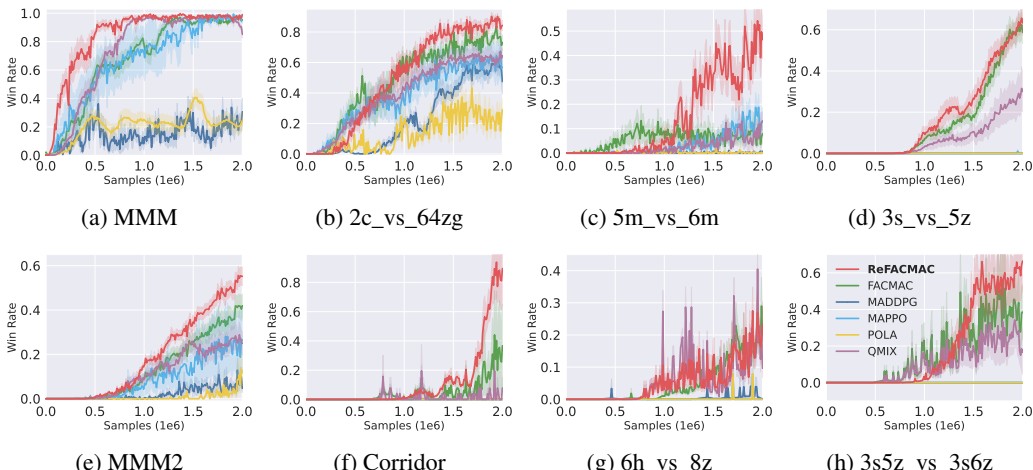

Figure 3: Mean win rate and standard error of ReFACMAC and baselines on SMAC maps across 10 seeds.

Figure 4 compares the performance of ReFACMAC against baselines on four selected MAMuJoCo environments. In all four environments, ReFACMAC achieves superior performance by the end of training and tends to reach its peak performance earlier. In Ant 2x4 (Figure 4c), learning a solution is difficult due to the asymmetric positioning of the agents which control opposing diagonal halves of a 4-legged agent. ReFACMAC and FACMAC both achieve the same maximum performance during training, but only ReFACMAC maintains an advantage despite the instability of the problem. ReFACMAC also achieves the *only* policy in Walker 2x3 (Figure 4d) which solves the task, obtaining SOTA performance on this very difficult control benchmark.

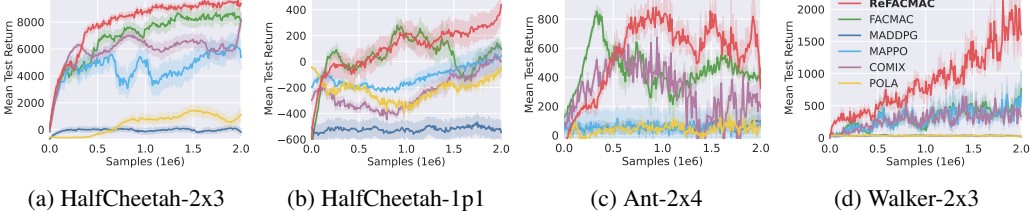

Figure 4: Mean performance and standard error of ReFACMAC and baselines on four Multi-Agent MuJoCo environments across 10 seeds.

## 6 HIGHER RECURSIVE REASONING

Thus far, we have considered practical implementations of recursive reasoning with policy gradient algorithms. While the results for ReMAPPO and ReFACMAC generally exhibit competitive sample-efficiency with just one level of recursive reasoning, our framework allows for further levels of recursion.

In the framework of higher recursive reasoning, we consider the application of Equation 2 an arbitrary number of times within one update step; that is, we repeatedly refine the policy gradient using the updates of the non-self agents at the previous recursion level. Note that at each recursion level, no new environment data is being collected - the policy gradient is being refined using the action distributions alone. We represent the parameters of agent $i$ after $k$ repeated recursions by $\boldsymbol{\theta}_i^{(k)}$. We define the pre-update parameters to be recursion level $0$. Thus, the vanilla update parameters denoted $\boldsymbol{\theta}_i'$ in Equation 1 would be represented by $\boldsymbol{\theta}_i^{(1)}$, and the recursive reasoning parameters $\boldsymbol{\theta}_i^{Re}$ in Equation 2 would be represented by $\boldsymbol{\theta}_i^{(2)}$. Figure 5 further illustrates our higher recursion framework.

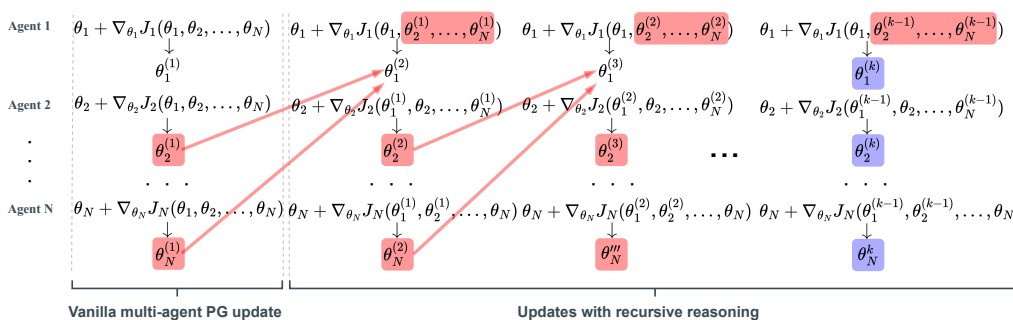

Figure 5: Higher recursive reasoning policy gradient updates for N agents. The non-self parameters used for recursive updates of Agent 1 are red and the final parameters of all agents after $k$ steps of recursion are blue.

### 6.1 THEORETICAL STUDY

In this section, we present a theoretical study which shows that repeated recursive reasoning with policy gradients converges monotonically to a local Nash equilibrium under certain conditions with finite iterates. Firstly, we demonstrate how an unbiased, infinite application of recursive reasoning leads to perfect anticipation of other agents' future strategies.

**Assumption 6.1.** *The gradient* $\nabla_{\boldsymbol{\theta}_i} J_i(\boldsymbol{\theta}_i, \boldsymbol{\theta}_{-i})$ *is* $L_i$-*Lipschitz with respect to* $\boldsymbol{\theta}_{-i}$, *i.e.*

$$\|\nabla_{\boldsymbol{\theta}_i} J_i(\boldsymbol{\theta}_{i,}, \boldsymbol{\theta}_{-i,1}) - \nabla_{\boldsymbol{\theta}_i} J_i(\boldsymbol{\theta}_{i,}, \boldsymbol{\theta}_{-i,2})\| \leq L_i\|\boldsymbol{\theta}_{-i,1} - \boldsymbol{\theta}_{-i,2}\|, \forall \boldsymbol{\theta}_i, \boldsymbol{\theta}_{-i}, \tag{9}$$

where $\boldsymbol{\theta}_{-i,1}$ and $\boldsymbol{\theta}_{-i,2}$ are two arbitrary points in the joint parameter space of the non-self agents. We define the maximum objective function Lipschitzness $L := \max_i\{L_i\}$ and the maximum agent learning rate $\eta := \max_i\{\eta_i\}$. We define the maximum objective function gradient across all agent parameters and objective functions $\nabla_{max} := \max_{i,\boldsymbol{\theta}_i,\boldsymbol{\theta}_{-i}} \|\nabla_{\boldsymbol{\theta}_i} J_i(\boldsymbol{\theta}_i, \boldsymbol{\theta}_{-i})\|$.

**Theorem 6.2.** *Suppose Assumption 6.1 holds. Then, the update step at the $k$'th level of reasoning is bounded:*

$$\|\boldsymbol{\theta}^{(k)} - \boldsymbol{\theta}^{(k-1)}\| \leq \eta(\eta L)^{k-1} N(N-1)^{k-1} \nabla_{max}. \tag{10}$$

*Assume the maximum learning rate $\eta$ satisfies $\eta < \frac{1}{L(N-1)}$. Then, the sequence $\{\boldsymbol{\theta}^{(k)}\}_{k=0}^{\infty}$ is a convergent sequence. Since $\boldsymbol{\theta}$ exists in a complete subspace of $\mathbb{R}^{\sum_i d_i}$, the convergent sequence $\{\boldsymbol{\theta}^{(k)}\}_{k=0}^{\infty}$ is Cauchy, i.e.,*

$$\exists C \in \mathbb{N} : \forall \epsilon > 0, (a > b > C \implies \|\boldsymbol{\theta}^{(a)} - \boldsymbol{\theta}^{(b)}\| < \epsilon). \tag{11}$$

*Since every Cauchy sequence has a limit, we denote the limit of $\{\boldsymbol{\theta}^{(k)}\}_{k=0}^{\infty}$ as $\lim_{k\to\infty} \boldsymbol{\theta}^{(k)} = \boldsymbol{\theta}^{(\infty)}$.*

According to Theorem 6.2, applying the recursive update with $k=\infty$ defines the following implicit algorithm:

$$\boldsymbol{\theta}_i^{(\infty)} \leftarrow \boldsymbol{\theta}_i + \eta_i \nabla_{\boldsymbol{\theta}_i} J_i\left(\boldsymbol{\theta}_i, \boldsymbol{\theta}_{-i}^{(\infty)}\right), \forall i \in \mathcal{I}, \tag{12}$$

which we denote the Generalized Semi-Proximal Point Method (GSPPM). The implication of the GSPPM is that the update of each agent responds exactly to the updated strategies of the other agents, maintaining mutual consistency.

Following from Theorem 6.2, we show that the convergence of GSPPM iterates in a non-convex non-concave strategy space can be analyzed via the game Jacobian around a local stationary point:

**Theorem 6.3.** *Let $\boldsymbol{\theta}^*$ be a stationary point in an N-player general sum game. This stationary point is a local Nash equilibrium, i.e. a point at which no agent's objective function has a non-zero gradient under a unilateral change in policy. Let $\boldsymbol{\eta}$ be a block matrix of the agent learning rates $\eta_i$. Let the components of the Hessian of each objective function at $\boldsymbol{\theta}^*$ be denoted*

$$\begin{pmatrix} \nabla^2_{\boldsymbol{\theta}_i \boldsymbol{\theta}_i} J_i(\boldsymbol{\theta}_i^* \boldsymbol{\theta}_{-i}^*) & \nabla^2_{\boldsymbol{\theta}_i \boldsymbol{\theta}_{-i}} J_i(\boldsymbol{\theta}_i^* \boldsymbol{\theta}_{-i}^*) \\ \nabla^2_{\boldsymbol{\theta}_{-i} \boldsymbol{\theta}_i} J_i(\boldsymbol{\theta}_i^* \boldsymbol{\theta}_{-i}^*) & \nabla^2_{\boldsymbol{\theta}_{-i} \boldsymbol{\theta}_{-i}} J_i(\boldsymbol{\theta}_i^* \boldsymbol{\theta}_{-i}^*) \end{pmatrix} = \begin{pmatrix} \boldsymbol{A}_i & \boldsymbol{B}_i \\ \boldsymbol{B}_i^\top & \boldsymbol{C}_i \end{pmatrix}.$$

*Furthermore, let $\boldsymbol{A}$ be the diagonal block matrix of all $\boldsymbol{A}_i$ matrices and $\boldsymbol{B}$ be the diagonal block matrix of all $\boldsymbol{B}_i$ matrices: $\boldsymbol{A} = \mathrm{diag}(\boldsymbol{A}_1, \ldots, \boldsymbol{A}_n), \boldsymbol{B} = \mathrm{diag}(\boldsymbol{B}_1, \ldots, \boldsymbol{B}_n)$. Let $\boldsymbol{D}$ be a complement-selection matrix for each set of agent parameters $\boldsymbol{\theta}_i$ such that $\boldsymbol{D}\boldsymbol{\theta} = [\boldsymbol{\theta}_{-1}, \ldots, \boldsymbol{\theta}_{-n}]^\top$*

*Suppose $\eta < \frac{1}{L(N-1)}$ such that the GSPPM iterates $\{\boldsymbol{\theta}_t^{(k)}\}_{k=0}^\infty$ form a Cauchy sequence. Let $\underset{max}{\lambda}(\boldsymbol{Z})$ and $\underset{min}{\lambda}(\boldsymbol{Z})$ refer to the maximum and minimum eigenvalues of a matrix $\boldsymbol{Z}$ respectively. Then, there exists a neighborhood $\mathcal{U} \in \mathbb{R}^{\sum_i d_i}$ around $\boldsymbol{\theta}^*$ such that if GSPPM starts in $\mathcal{U}$, the iterates $\{\boldsymbol{\theta}_t^{(k)}\}_{k=0}^\infty$ satisfy:*

$$\|\boldsymbol{\theta}^{(\infty)} - \boldsymbol{\theta}^*\| \le \frac{\underset{max}{\lambda}(I + \boldsymbol{\eta}\boldsymbol{A})^2}{\underset{min}{\lambda}(I - \boldsymbol{\eta}\boldsymbol{B}\boldsymbol{D})^2} \|\boldsymbol{\theta} - \boldsymbol{\theta}^*\|. \tag{13}$$

*Moreover, for any $\boldsymbol{\eta}$ satisfying $\frac{\underset{max}{\lambda}(I + \boldsymbol{\eta}\boldsymbol{A})^2}{\underset{min}{\lambda}(I - \boldsymbol{\eta}\boldsymbol{B}\boldsymbol{D})^2} < 1$, the iterates converge asymptotically to $\boldsymbol{\theta}^*$. Hence, GSPPM iterates reach an $\epsilon$-Nash equilibrium.*

**Theorem 6.4.** *Suppose the conditions of Theorem 6.3 apply. Then, the finite iterates $\{\boldsymbol{\theta}_t^{(k)}\}$ satisfy:*

$$\|\boldsymbol{\theta}^{(k)} - \boldsymbol{\theta}^*\|^2 \le \left( \underset{max}{\lambda}(I + \boldsymbol{\eta}\boldsymbol{A})^2 + 2\underset{max}{\lambda}(I + \boldsymbol{\eta}\boldsymbol{A})\underset{max}{\lambda}(\boldsymbol{\eta}\boldsymbol{B}\boldsymbol{D}) \right) \|\boldsymbol{\theta} - \boldsymbol{\theta}^*\|^2 \tag{14}$$

$$+ 2\underset{max}{\lambda}(I + \boldsymbol{\eta}\boldsymbol{A})\underset{max}{\lambda}(\boldsymbol{\eta}\boldsymbol{B}\boldsymbol{D})(\|\boldsymbol{\theta} - \boldsymbol{\theta}^*\|\nabla_{max}) \tag{15}$$

$$+ \underset{max}{\lambda}(\boldsymbol{\eta}\boldsymbol{B}\boldsymbol{D})^2 \|\boldsymbol{\theta}^{(k-1)} - \boldsymbol{\theta}^*\|^2. \tag{16}$$

*Moreover, for any $\boldsymbol{\eta}$ satisfying $\underset{max}{\lambda}(\boldsymbol{\eta}\boldsymbol{B}\boldsymbol{D})^2 < 1$, the finite iterates converge asymptotically to $\boldsymbol{\theta}^*$.*

## 6.2 ILLUSTRATIVE EXAMPLE

We further examine higher recursive reasoning using a didactic example of a simple cooperative game with two point agents taking continuous actions in a 2D space. The agents have one parameter each and produce a one-dimensional action (the angle of their next move). The highest reward is achieved when the chosen direction of each agent points towards the *future* location of the other agent. Assuming the agents move kinematically, the optimal solution is for the agents to move towards each other in a straight line (see Appendix B for details). However, the naive policy update for this game under fictitious play is for each agent to choose its next action to intercept with the *previous* action of the other agent. The top two figures of Figure 6a illustrate this problem: each agent's new action (red arrows) points towards the destination of the other agent under the other agent's old policy (yellow arrows). Hence, the naive update leads to a lack of mutual consistency.

Figure 6a bottom left shows the policy update after 2 levels of recursion: now the agents update to intercept the other agent *after* the other agent's naive policy update, resulting in better coordination. As the number of recursions increases, the policies converge on an $\epsilon$-bound of the optimal solution. Figure 6b shows the progression of the agent parameters with gradient ascent and momentum; we use the true gradient and objective function $J(\boldsymbol{\theta}_1, \boldsymbol{\theta}_2)$. The benefits of recursive reasoning are evinced by the fact that increasing $k$-levels (darker points) exhibit monotonic convergence to the optimal parameters $\boldsymbol{\theta}_1^*, \boldsymbol{\theta}_2^*$ at each update step, as supported by Theorem 6.4. Figure 7 shows the distance from the Nash equilibrium for one update step near the stationary point in Figure 6b.

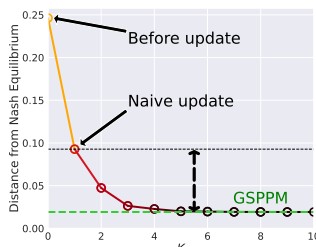

Figure 7: Convergence to GSPPM with higher recursive reasoning.

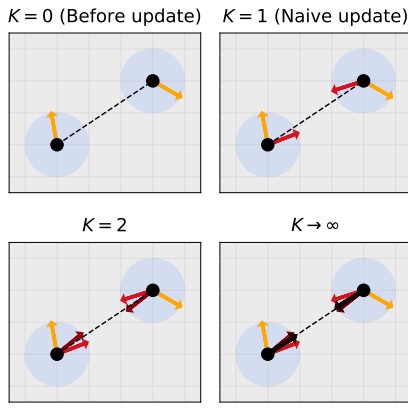 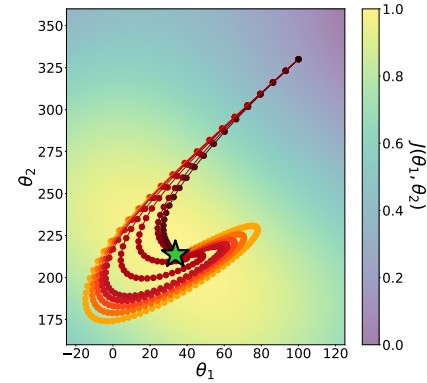

(a) Agents taking continuous actions with increasing recursive reasoning levels (darkening arrows) converging on the optimal actions (dashed line).

(b) Gradient ascent to the optimal parameters (green star) with recursion (darker colors for higher recursive reasoning levels).

Figure 6: An illustrative continuous cooperative game with two point agents using recursive reasoning.

### 6.3 HIGHER RECURSIVE REASONING IN DEEP MULTI-AGENT REINFORCEMENT LEARNING

Consolidating with the theoretical study in Section 6.1, we compare the sample efficiency of higher recursive reasoning in ReFACMAC and ReMADDPG up to $k=5$ in HalfCheetah-2x3 (MAMuJoCo) and MMM (SMAC) in Figure 8. While higher recursion exhibits stable performance increases, most of the benefits appear to materialize by $k=2$ in both the ablation and didactic example in Section 6.2. Similar lookahead methods in optimizers (Suh & Ma, 2025) and GANs (Liu & Pavel, 2022) exhibit polynomial convergence rates, achieving the majority of benefits within the first few iterates. We leave an analysis of the convergence rates of higher recursive reasoning with finite data to future investigations. Interestingly, Figure 8c demonstrates a situation where recursions above $k=2$ find a better policy mode.

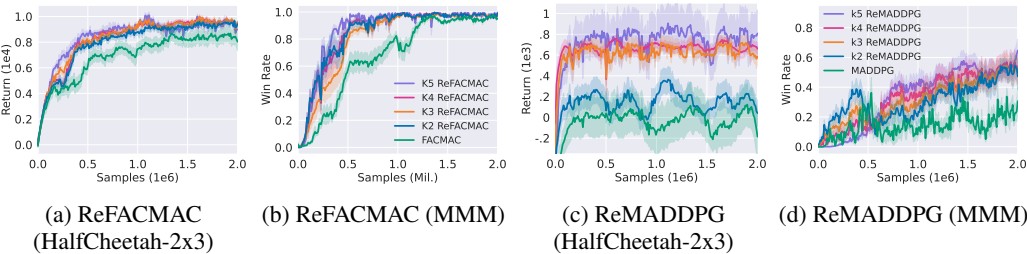

(a) ReFACMAC (HalfCheetah-2x3)

(b) ReFACMAC (MMM)

(c) ReMADDPG (HalfCheetah-2x3)

(d) ReMADDPG (MMM)

Figure 8: Ablations for higher recursive reasoning with ReFACMAC and ReMADDPG on SMAC (MMM) and MAMuJoCo (HalfCheetah-2x3).

## 7 CONCLUSION

We present a framework for recursive reasoning for multi-agent policy gradient algorithms. We introduce our general recursive formulation and practically realize it in both on and off-policy regimes, achieving SOTA performance and sample efficiency against competitive baselines in challenging MARL benchmarks. We show that the recursive reasoning paradigm can be extended, and prove theoretical convergence properties of finite and infinite recursive iterates with respect to local equilibria. We leave it to future work to understand the convergence rates of recursive iterates in the finite data setting. We also hope to address the assumption of access to non-self agent policy distributions, which could be modeled to similar effect.

## 8 REPRODUCIBILITY STATEMENT

To ensure reproducibility of our findings, we provide details of benchmark configurations, algorithm hyperparameters, architectures, and other training techniques and procedures in the Appendix. Our theoretically contributions can be verified by proofs provided in the Appendix. We provide JAX and PyTorch implementations of our algorithms as supplementary material.

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

# A PROOFS

## A.1 PROOF OF THEOREM 6.2

Recall Assumption 6.1. We analyze the pattern in successive updates of $\boldsymbol{\theta}$ as $k$ increases.

Consider level $k=1$:

$$\boldsymbol{\theta}_i^{(1)} = \boldsymbol{\theta}_i + \eta_i \nabla_{\boldsymbol{\theta}_i} J(\boldsymbol{\theta}_i, \boldsymbol{\theta}_{-i}) \quad \forall i \in \mathcal{I}. \tag{17}$$

The jump between $\boldsymbol{\theta}_i^{(1)}$ and $\boldsymbol{\theta}_i$ is

$$\|\boldsymbol{\theta}_i^{(1)} - \boldsymbol{\theta}_i\| = \eta_i \|\nabla_{\boldsymbol{\theta}_i} J_i(\boldsymbol{\theta}_i, \boldsymbol{\theta}_{-i})\| \quad \forall i \in \mathcal{I}. \tag{18}$$

Thus, for all agents

$$
\begin{aligned}
\|\boldsymbol{\theta}^{(1)} - \boldsymbol{\theta}\| &\leq \sum_{i=1}^{N} \|\boldsymbol{\theta}_i^{(1)} - \boldsymbol{\theta}_i\| = \sum_{i=1}^{N} \eta_i \|\nabla_{\boldsymbol{\theta}_i} J_i(\boldsymbol{\theta}_i, \boldsymbol{\theta}_{-i})\| \\
&\leq \eta \sum_{i=1}^{N} \|\nabla_{\boldsymbol{\theta}_i} J_i(\boldsymbol{\theta}_i, \boldsymbol{\theta}_{-i})\| \\
&\leq \eta N \max_i \|\nabla_{\boldsymbol{\theta}_i} J_i(\boldsymbol{\theta}_i, \boldsymbol{\theta}_{-i})\| \\
&\leq \eta N \nabla_{\max}.
\end{aligned}
\tag{19}
$$

Now consider level $k=2$:

$$\boldsymbol{\theta}_i^{(2)} = \boldsymbol{\theta}_i + \eta_i \nabla_{\boldsymbol{\theta}_i} J(\boldsymbol{\theta}_i, \boldsymbol{\theta}_{-i}^{(1)}) \quad \forall i \in \mathcal{I}. \tag{20}$$

The jump between $\boldsymbol{\theta}_t^{(2)}$ and $\boldsymbol{\theta}^{(1)}$ is

$$
\begin{aligned}
\|\boldsymbol{\theta}_i^{(2)} - \boldsymbol{\theta}_i^{(1)}\| &= \|\boldsymbol{\theta}_i + \eta_i \nabla_{\boldsymbol{\theta}_i} J_i(\boldsymbol{\theta}_i, \boldsymbol{\theta}_{-i}^{(1)}) - \boldsymbol{\theta}_i - \eta_i \nabla_{\boldsymbol{\theta}_i} J_i(\boldsymbol{\theta}_i, \boldsymbol{\theta}_{-i})\| \\
&= \eta_i \|\nabla_{\boldsymbol{\theta}_i} J_i(\boldsymbol{\theta}_i, \boldsymbol{\theta}_{-i}^{(1)}) - \nabla_{\boldsymbol{\theta}_i} J_i(\boldsymbol{\theta}_i, \boldsymbol{\theta}_{-i})\| \\
&\leq \eta_i L_i \|\boldsymbol{\theta}_{-i}^{(1)} - \boldsymbol{\theta}_{-i}\| \quad \forall i \in \mathcal{I}.
\end{aligned}
\tag{21}
$$

Thus, for all agents

$$
\begin{aligned}
\|\boldsymbol{\theta}^{(2)} - \boldsymbol{\theta}^{(1)}\| &\leq \sum_{i=1}^{N} \|\boldsymbol{\theta}_i^{(2)} - \boldsymbol{\theta}_i^{(1)}\| \\
&\leq \sum_{i=1}^{N} \eta_i L_i \|\boldsymbol{\theta}_{-i}^{(1)} - \boldsymbol{\theta}_{-i}\| \\
&\leq \sum_{i=1}^{N} \eta_i L_i \sum_{j \neq i} \|\boldsymbol{\theta}_j^{(1)} - \boldsymbol{\theta}_j\| \\
&= \sum_{i=1}^{N} \eta_i L_i \sum_{j \neq i} \eta_j \|\nabla_{\boldsymbol{\theta}_j} J_j(\boldsymbol{\theta}_j, \boldsymbol{\theta}_{-j})\| \\
&\leq \sum_{i=1}^{N} \eta_i L_i (N-1) \eta \nabla_{max} \\
&\leq \eta^2 L N (N-1) \nabla_{max}.
\end{aligned}
\tag{22}
$$

Now consider level $k=3$:

$$\boldsymbol{\theta}_i^{(3)} = \boldsymbol{\theta}_i + \eta_i \nabla_{\boldsymbol{\theta}_i} J(\boldsymbol{\theta}_i, \boldsymbol{\theta}_{-i}^{(2)}) \quad \forall i \in \mathcal{I}. \tag{23}$$

The jump between $\boldsymbol{\theta}_t^{(3)}$ and $\boldsymbol{\theta}^{(2)}$ is

$$\begin{aligned}
\|\boldsymbol{\theta}_i^{(3)} - \boldsymbol{\theta}_i^{(2)}\| &= \eta_i \|\nabla_{\boldsymbol{\theta}_i} J_i(\boldsymbol{\theta}_i, \boldsymbol{\theta}_{-i}^{(2)}) - \nabla_{\boldsymbol{\theta}_i} J_i(\boldsymbol{\theta}_i, \boldsymbol{\theta}_{-i}^{(1)})\| \\
&\leq \eta_i L_i \|\boldsymbol{\theta}_{-i}^{(2)} - \boldsymbol{\theta}_{-i}^{(1)}\| \quad \forall i \in \mathcal{I}.
\end{aligned} \tag{24}$$

Thus, for all agents

$$\begin{aligned}
\|\boldsymbol{\theta}^{(3)} - \boldsymbol{\theta}^{(2)}\| &\leq \sum_{i=1}^{N} \|\boldsymbol{\theta}_i^{(3)} - \boldsymbol{\theta}_i^{(2)}\| \\
&\leq \sum_{i=1}^{N} \eta_i L_i \|\boldsymbol{\theta}_i^{(2)} - \boldsymbol{\theta}_i^{(1)}\| \\
&\leq \sum_{i=1}^{N} \eta_i L_i \sum_{j \neq i} \|\boldsymbol{\theta}_j^{(2)} - \boldsymbol{\theta}_j^{(1)}\| \\
&\leq \sum_{i=1}^{N} \eta_i L_i \sum_{j \neq i} \eta_j L_j \|\boldsymbol{\theta}_{-j}^{(1)} - \boldsymbol{\theta}_{-j}\| \\
&\leq \sum_{i=1}^{N} \eta_i L_i \sum_{j \neq i} \eta_j L_j \sum_{l \neq j} \|\boldsymbol{\theta}_m^{(1)} - \boldsymbol{\theta}_m\| \\
&= \sum_{i=1}^{N} \eta_i L_i \sum_{j \neq i} \eta_j L_j \sum_{l \neq j} \eta_m \|\nabla_{\boldsymbol{\theta}_m} J_m(\boldsymbol{\theta}_m, \boldsymbol{\theta}_{-m})\| \\
&= \sum_{i=1}^{N} \eta_i L_i \sum_{j \neq i} \eta_j L_j \sum_{l \neq j} \eta_m \|\nabla_{\boldsymbol{\theta}_m} J_m(\boldsymbol{\theta}_m, \boldsymbol{\theta}_{-m})\| \\
&\leq \eta^3 L^2 N(N-1)^2 \nabla_{max}.
\end{aligned} \tag{25}$$

We see by induction that any consecutive states during the recursive procedure are bounded by

$$\|\boldsymbol{\theta}^k - \boldsymbol{\theta}^{k-1}\| \leq \eta(\eta L)^{k-1} N(N-1)^{k-1} \nabla_{max}. \tag{26}$$

Let $\eta < \frac{1}{L(n-1)}$ such that the difference between between two recursive steps is a contraction. Consider the difference $\|\boldsymbol{\theta}^{(a)} - \boldsymbol{\theta}^{(b)}\|$, where $a > b > 0$:

$$\|\boldsymbol{\theta}^{(a)} - \boldsymbol{\theta}^{(b)}\| = \| \sum_{j=b+1}^{a} \left( \boldsymbol{\theta}^{(j)} - \boldsymbol{\theta}^{(j-1)} \right) \|$$

$$\leq \sum_{j=b+1}^{a} \|\boldsymbol{\theta}^{(j)} - \boldsymbol{\theta}^{(j-1)}\|$$

$$\leq \sum_{j=b+1}^{a} \eta(\eta L)^{j-1} N (N-1)^{j-1} \nabla_{max} \tag{27}$$

$$\leq N\nabla_{max} \left( \sum_{j=b+1}^{a} (N-1)^{j-1} L^{j-1} \right) \eta \left( \sum_{j=b+1}^{a} \eta^{j-1} \right)$$

$$\leq N\nabla_{max} \left( \sum_{j=b+1}^{a} (N-1)^{j-1} L^{j-1} \right) \eta \left( \sum_{j=b+1}^{a} \eta^{b-1} \right) \approx \mathcal{O}(\eta^b).$$

Thus, for any $\epsilon > 0$, we can solve for b such that $\eta(\eta L)^{k-1} N(N-1)^{k-1} \nabla_{max} < \epsilon$, or

$$\exists C \in \mathbb{N} : \forall \epsilon > 0, (a > b > C \implies \|\boldsymbol{\theta}^{(a)} - \boldsymbol{\theta}^{(b)}\| < \epsilon). \tag{28}$$

Hence $\{\boldsymbol{\theta}^{(k)}\}_{k=0}^{\infty}$ is a Cauchy sequence. Since $\boldsymbol{\theta}$ lies in a complete subspace of $\mathbb{R}^{\sum_i d_i}$, the Cauchy sequence has a limit: $\lim_{k \to \infty} \boldsymbol{\theta}^{(k)} = \boldsymbol{\theta}^{\infty}$. $\qquad\square$

### A.2 PROOF OF THEOREM 6.3

We abuse notation by denoting agent $i$'s parameters at update step $t$ with $\boldsymbol{\theta}_{t,i}$ and the parameters one update step later at $t + 1$ with $\boldsymbol{\theta}_{t+1,i}$, where an arbitrary number of recursive reasoning steps were taken in between. Let us define $\hat{\boldsymbol{\theta}}_{t,i} = \boldsymbol{\theta}_{t,i} - \boldsymbol{\theta}_i^*$ and $\hat{\boldsymbol{\theta}}_t = [\hat{\boldsymbol{\theta}}_{t,1}, ... \hat{\boldsymbol{\theta}}_{t,N}]^T$ for all $i \in \mathcal{I}$. It follows by linearizing the system about the stationary point $\boldsymbol{\theta}^*$,

$$\hat{\boldsymbol{\theta}}_{t+1,i} = \boldsymbol{\theta}_{t,i} + \eta_i \nabla_{\boldsymbol{\theta}_i} J_i(\boldsymbol{\theta}_{t,i} \boldsymbol{\theta}_{t,-i}) - \boldsymbol{\theta}^*$$

$$\approx \left( I + \eta_i \nabla^2_{\boldsymbol{\theta}_i, \boldsymbol{\theta}_i} J_i(\boldsymbol{\theta}_i^*, \boldsymbol{\theta}_{-i}^*), \eta_i \nabla^2_{\boldsymbol{\theta}_i, \boldsymbol{\theta}_{-i}} J_i(\boldsymbol{\theta}_i^*, \boldsymbol{\theta}_{-i}^*) \right) \begin{pmatrix} \hat{\boldsymbol{\theta}}_{t,i} \\ \hat{\boldsymbol{\theta}}_{t+1,-i} \end{pmatrix} \quad \textit{First order Taylor expansion}$$

$$= \hat{\boldsymbol{\theta}}_{t,i} + \eta_i \boldsymbol{A}_i \hat{\boldsymbol{\theta}}_{t,i} + \eta_i \boldsymbol{B}_i \hat{\boldsymbol{\theta}}_{t+1,-i}$$

$$\therefore \hat{\boldsymbol{\theta}}_{t+1} = \hat{\boldsymbol{\theta}}_t + \boldsymbol{\eta} \boldsymbol{A} \hat{\boldsymbol{\theta}}_t + \boldsymbol{\eta} \boldsymbol{B} \boldsymbol{D} \hat{\boldsymbol{\theta}}_{t+1}. \tag{29}$$

By analyzing the distance $r$ of the GSPPM iterates from the stationary point,

$$r_{t+1}^2 = \|\hat{\boldsymbol{\theta}}_{t+1}\|^2$$

$$= \hat{\boldsymbol{\theta}}_t^T (I + \boldsymbol{\eta} \boldsymbol{A})^T (I - \boldsymbol{\eta} \boldsymbol{B} \boldsymbol{D})^{-T} (I - \boldsymbol{\eta} \boldsymbol{B} \boldsymbol{D})^{-1} (I + \boldsymbol{\eta} \boldsymbol{A}) \hat{\boldsymbol{\theta}}_t \tag{30}$$

$$\leq \frac{\lambda_{max} (I + \boldsymbol{\eta} \boldsymbol{A})^2}{\lambda_{min} (I - \boldsymbol{\eta} \boldsymbol{B} \boldsymbol{D})^2} r_t^2.$$

Thus, for any $\{\eta_i\}$ satisfying $\frac{\lambda_{max}(I+\boldsymbol{\eta} \boldsymbol{A})^2}{\lambda_{min}(I-\boldsymbol{\eta} \boldsymbol{B} \boldsymbol{D})^2} < 1$, GSPPM iterates converge asymptotically to the local Nash equilibrium. $\qquad\square$

### A.3 PROOF OF THEOREM 6.4

Let us define $\hat{\boldsymbol{\theta}}_{t,i}^{(k)} = \boldsymbol{\theta}_{t,i}^{(k)} - \boldsymbol{\theta}_{t,i}^*$ and $\hat{\boldsymbol{\theta}}^{(k)} = [\hat{\boldsymbol{\theta}}_1^{(k)}, ... \hat{\boldsymbol{\theta}}_n^{(k)}]^T$ for all $i \in \mathcal{I}$. It follows by linearizing the system about the stationary point $\boldsymbol{\theta}^*$,

$$
\begin{aligned}
\hat{\boldsymbol{\theta}}_{t+1,i}^{(k)} &= \boldsymbol{\theta}_{t,i} + \eta_i \nabla_{\boldsymbol{\theta}_{t,i}} J_i(\boldsymbol{\theta}_{t,i} \boldsymbol{\theta}_{t,-i}^{(k-1)}) - \boldsymbol{\theta}_i^* \\
&\approx \left(I + \eta_i \nabla_{\boldsymbol{\theta}_i, \boldsymbol{\theta}_i}^2 J_i(\boldsymbol{\theta}_{t,i}^*, \boldsymbol{\theta}_{t,-i}^*), \eta_i \nabla_{\boldsymbol{\theta}_i, \boldsymbol{\theta}_{-i}}^2 J_i(\boldsymbol{\theta}_{t,i}^*, \boldsymbol{\theta}_{t,-i}^*)\right) \begin{pmatrix} \hat{\boldsymbol{\theta}}_{t,i} \\ \hat{\boldsymbol{\theta}}_{t,-i}^{(k-1)} \end{pmatrix} \quad \text{First order Taylor expansion} \\
&= \hat{\boldsymbol{\theta}}_{t,i} + \eta_i \boldsymbol{A}_i \hat{\boldsymbol{\theta}}_{t,i} + \eta_i \boldsymbol{B}_i \hat{\boldsymbol{\theta}}_{t,-i}^{(k-1)} \\
\therefore \hat{\boldsymbol{\theta}}_t^{(k)} &= \hat{\boldsymbol{\theta}}_t + \boldsymbol{\eta} \boldsymbol{A} \hat{\boldsymbol{\theta}}_t + \boldsymbol{\eta} \boldsymbol{B} \boldsymbol{D} \hat{\boldsymbol{\theta}}_t^{(k-1)}.
\end{aligned}
\tag{31}
$$

By analyzing the distance $r^{(k)}$ of the iterates from the stationary point,

$$
\begin{aligned}
\left(r_t^{(k)}\right)^2 &= \|\hat{\boldsymbol{\theta}}_t^{(k)}\|^2 \\
&= \hat{\boldsymbol{\theta}}_t^T (I + \boldsymbol{\eta} \boldsymbol{A})^T (I + \boldsymbol{\eta} \boldsymbol{A}) \hat{\boldsymbol{\theta}}_t + \hat{\boldsymbol{\theta}}_t^T (I + \boldsymbol{\eta} \boldsymbol{A})^T \boldsymbol{\eta} \boldsymbol{B} \boldsymbol{D} \hat{\boldsymbol{\theta}}_t^{(k-1)} \\
&\quad + (\hat{\boldsymbol{\theta}}_t^{(k-1)})^T \boldsymbol{D}^T \boldsymbol{B}^T \boldsymbol{\eta}^T (I + \boldsymbol{\eta} \boldsymbol{A}) \hat{\boldsymbol{\theta}}_t + (\hat{\boldsymbol{\theta}}_t^{(k-1)})^T \boldsymbol{D}^T \boldsymbol{B}^T \boldsymbol{\eta}^T \boldsymbol{\eta} \boldsymbol{B} \boldsymbol{D} \hat{\boldsymbol{\theta}}_t^{(k-1)} \\
&\leq \left(\underset{max}{\lambda} (I + \boldsymbol{\eta} \boldsymbol{A})^2 + 2 \underset{max}{\lambda} (I + \boldsymbol{\eta} \boldsymbol{A}) \underset{max}{\lambda} (\boldsymbol{\eta} \boldsymbol{B} \boldsymbol{D})\right) \left(r_t^{(0)}\right)^2 + \\
&\quad 2 \underset{max}{\lambda} (I + \boldsymbol{\eta} \boldsymbol{A}) \underset{max}{\lambda} (\boldsymbol{\eta} \boldsymbol{B} \boldsymbol{D}) \left(r_t^{(0)} \nabla_{max}\right) + \underset{max}{\lambda} (\boldsymbol{\eta} \boldsymbol{B} \boldsymbol{D})^2 \left(r_t^{(k-1)}\right)^2.
\end{aligned}
\tag{32}
$$

defining the bound of the finite-k iterates to the stationary point $\hat{\boldsymbol{\theta}}^*$.

Hence, for any $\{\eta_i\}$ satisfying $\underset{max}{\lambda} (\boldsymbol{\eta} \boldsymbol{B} \boldsymbol{D})^2 < 1$, the iterates converge asymptotically to the local Nash Equilibrium. $\qquad\square$

## B DETAILS OF THE ILLUSTRATIVE EXAMPLE

We report the full details of the toy problem introduced in Sec. 6.2, which we here refer to as the *Meet-up* problem.

**Environment properties.** The problem is designed as a simple 2-player continuous cooperative game in a 2D space. The state of the game $s = (s_1, s_2) \in \mathbb{R}^4$ encodes the location of the two players, with $s_i \in \mathbb{R}^2$. For the sake of simplicity, agents can only move by a fixed distance step of 1 around their current position, towards a chosen direction. The initial state of the two agents is deterministic and fixed to $\iota = (\iota_1 = (0,0), \iota_2 = (3,2))$. We assume undiscounted returns ($\gamma = 1$) and terminate an episode when the agents effectively meet each other as a result of their actions.

**Policy parameterization.** Although one-dimensional continuous actions are trivially tractable for one-step games, sequential decision making problems demand finding policies that respond optimally for any possible configuration $s$ of the game. Here, we reduce the complexity of the problem by conveniently parameterizing each agent as single-parameter policies. In particular, we define an agent action as a 1-DoF unit vector $a_i \in \mathbb{R}^2$, and parameterize the deterministic policy of agent $i$ with $\theta_i \in \mathbb{R}$, as

$$
\pi_i(s) = \begin{cases} (\cos \theta_i, \sin \theta_i)^\top & \text{if, } s = \iota \\ \pi_i^*(s) & \text{if, } s \neq \iota \end{cases}
\tag{33}
$$

where, $\pi_i^*(s) = \frac{s_{-i} - s_i}{\|s_{-i} - s_i\|}$ is the optimal policy that goes straight towards the other agent. In other words, we assume that both agents will act optimally after taking the first action, and we only

parametrize the agents decisions at the starting state. This design choice allows to easily study the joint policy space directly, as well as computing the closed-form solution of the return $J(\cdot)$ (see below).

**Solving the Meet-up problem.** We design the reward function of the Meet-up problem to reward each agent for getting closer to the other agent after the effect of both actions. We achieve this by computing the cosine similarity between the agent's action $a_i$ and the actual direction that would have led closest to the other agent:

$$R_i(s, a, s') = a_i \cdot \pi_i^*(s_i, s'_{-i}) - 1 = a_i \cdot \frac{s'_{-i} - s_i}{\|s'_{-i} - s_i\|} - 1. \tag{34}$$

Here, we denote the joint action as $a = (a_1, a_2)$, and the next state as $s' = (s_1 + a_1, s_2 + a_2)$. Note that a $-1$ offset is added so that both the reward signal and the return $J_i(\theta_i, \theta_{-i})$ of each agent is always $\leq 0$. In turn, this makes the computation of the optimal value function $V^*(s)$ of this game trivial: the strategy of moving towards each other in a straight line leads to returns of $0$ from any state $s$; since this is the maximum return, this joint policy must also be optimal, and $V^*(s) = 0 \; \forall s$ is the unique optimal value function. We now derive the analytical form of $J_i(\theta_i, \theta_{-i})$, as needed to compute the recursive gradient updates. Given that both agents are assumed to act optimally in any state besides the starting state, we can conveniently write the return as

$$\begin{aligned} J_i(\theta_i, \theta_{-i}) &= R_i(\iota, a, s') + V_i(s') \\ &= R_i(\iota, a, s') + V^*(s') = \\ &= R_i(\iota, a, s') \end{aligned} \tag{35}$$

where $s'$ is the resulting state after the players' first actions $a_1 = (\cos\theta_1, \sin\theta_1)$ and $a_2 = (\cos\theta_2, \sin\theta_2)$. Following this, we may therefore compute the gradient of the return for any pair of agent policies $\theta_1, \theta_2$ in closed form:

$$\begin{aligned} \nabla_{\theta_i} J_i(\theta_i, \theta_{-i}) &= \nabla_{\theta_i} R_i(\iota, a, s') \\ &= \nabla_{\theta_i} \Big( a_i \cdot \pi_i^*(\iota_i, s'_{-i}) \Big) \\ &= \nabla_{\theta_i} \Big( a_i \Big) \cdot \pi_i^*(\iota_i, s'_{-i}) \\ &= \nabla_{\theta_i} \begin{pmatrix} \cos\theta_i \\ \sin\theta_i \end{pmatrix} \cdot \pi_i^*(\iota_i, s'_{-i}) \\ &= \begin{pmatrix} -\sin\theta_i \\ \cos\theta_i \end{pmatrix} \cdot \pi_i^*(\iota_i, s'_{-i}) \end{aligned} \tag{36}$$

In conclusion, Eq. 36 allows us to compute the recursive reasoning steps with the true analytical gradient of the return.

## C   PRACTICAL IMPLEMENTATION OF THE RECURSIVE POLICY GRADIENT IN REMAPPO

**The performance difference lemma (PDL)** (Kakade & Langford, 2002) **can be extended to the recursive reasoning multi-agent setting** (Zhao et al., 2023; Li et al., 2022).

**Lemma C.1.** *It can be shown that for agent $i$,*

$$J_i(\pi_i, \boldsymbol{\pi}_{-i}^{(0)}) - J_i(\pi_i^{(0)}, \boldsymbol{\pi}_{-i}^{(0)}) = \frac{1}{1 - \gamma} \mathbb{E}_{(s, \boldsymbol{a}) \sim d^{\pi_i, \boldsymbol{\pi}_{-i}^{(0)}}, \pi_i, \boldsymbol{\pi}_{-i}^{(0)}} \left[ A_i^{\boldsymbol{\pi}^{(0)}}(s, \boldsymbol{a}) \right] \tag{37}$$

$$= \frac{1}{1 - \gamma} \mathbb{E}_{(s, \boldsymbol{a}) \sim d^{\pi_i, \boldsymbol{\pi}_{-i}^{(0)}}, \pi_i^{(0)}, \boldsymbol{\pi}_{-i}^{(0)}} \left[ \frac{\pi_i}{\pi_i^{(0)}} A^{\boldsymbol{\pi}^{(0)}}(s, \boldsymbol{a}) \right]. \tag{38}$$

*This is the version of the PDL used by MAPPO; since it does not consider the update of the other agents, they are treated like part of the environment.*

*Now consider the PDL when updating recursively against non-self agents:*

$$J_i(\pi_i, \boldsymbol{\pi}_{-i}^{(k)}) - J_i(\pi_i^{(0)}, \boldsymbol{\pi}_{-i}^{(0)}) = \frac{1}{1-\gamma} \mathbb{E}_{(s,\boldsymbol{a}) \sim d^{\pi_i, \boldsymbol{\pi}_{-i}^{(k)}}, \pi_i, \boldsymbol{\pi}_{-i}^{(k)}} \left[ A_i^{\boldsymbol{\pi}^{(0)}}(s, \boldsymbol{a}) \right] \tag{39}$$

$$= \frac{1}{1-\gamma} \mathbb{E}_{(s,\boldsymbol{a}) \sim d^{\pi_i, \boldsymbol{\pi}_{-i}^{(0)}}, \pi_i^{(0)}, \boldsymbol{\pi}_{-i}^{(0)}} \left[ \frac{\pi_i}{\pi_i^{(0)}} \frac{\boldsymbol{\pi}_{-i}^{(k)}}{\boldsymbol{\pi}_{-i}^{(0)}} A^{\boldsymbol{\pi}^{(0)}}(s, \boldsymbol{a}) \right], \tag{40}$$

*which includes a correction ratio accounting for the change in action distribution of the recursively updated agents. Note that both cases feature a mismatch in the discounted state distribution $d$ of the PDL and the sample distribution ($d^{\pi_i, \boldsymbol{\pi}_{-i}}$); this mismatch is ignored in practice.*

## D   ENVIRONMENT DETAILS

All SMAX maps were configured according to the default settings seen in Rutherford et al. (2023). All experiments using SMAC mostly used the default team configurations, rewards, and observations as the SMAC benchmark (Samvelyan et al., 2019). The state space was modified slightly by including the last actions of each agent (using the inbuilt feature in the StarCraft II environment) as this was found to stabilize learning for all algorithms. All MAMuJoCo environments and agents are configured according to the default configurations used in Peng et al. (2021) where they were introduced. Each agent observes the positions of its own body parts, receives a common team reward that depends on the task, and controls only its joints. The exact configurations and rewards can be seen at `https://robotics.farama.org/envs/MaMuJoCo/`.

## E   EXPERIMENTAL DETAILS

### E.1   SMAX

Each baseline is run with the settings seen in Rutherford et al. (2023). Each baseline uses $1e7$ total training steps and is trained against the 'HeuristicEnemySMAX' AI, updated every 128 steps, and uses a $\gamma$ of 0.99. **Off-policy algorithms** (QMIX, VDN, IQL) are trained with 16 parallel environments with a buffer size of 5000 and a batch size of 32. Each uses Adam optimizers with a learning rate of $5e-5$, and performs $\epsilon$-greedy exploration during training time with and $\epsilon$ that decays from 1 to 0.05 over the first 10% of total steps (learning is also paused until the $\epsilon$ decay is concluded). Neural networks use a hidden size of 512 and relu activations, hard target updates every 10 updates, 8 update epochs, and a reward scale of 10 (the reward scale of the original SMAC environments). The maximum gradient norm is constrained to be 10. In QMIX, the mixer embedding dimension is 64, the mixer hypernet hidden dimension is 256, and the initial scale of the kernel weights of the mixer weights is set to 0.001. Baselines are evaluated every 5% of total steps for 128 steps across 512 environments. **On-policy algorithms** (ReMAPPO, MAPPO, IPPO, POLA) are trained with 64 parallel environments. Each uses Adam optimizers with a learning rate of $4e-3$ which is annealed to 0 over the entire course of training. Neural networks use a hidden size of 128 and relu activations, 2 minibatch updates, and 2 update epochs. The maximum gradient norm is constrained to be 0.5. The value of $\lambda$ for the GAE is set to 0.95, the value of $\epsilon$ for surrogate clipping is 0.2, the value loss coefficient is 0.5, and no entropy bonus is provided.

### E.2   MULTI-AGENT MUJOCO AND SMAC

For each algorithm, we evaluate the performance in the following manner: we pause training after 10,000 steps and run a fixed number of independent test episodes (10 for MAMuJoCo and 32 for SMAC). During these test episodes, each agent acts greedily in a decentralized fashion (DDPG agents don't use action noise, DQN agents don't select random actions, etc.). The mean performance of the agents is reported in MAMuJoCo (the performance for each agent is identical since they share a common objective) and the mean success rate is reported for SMAC. Note that we chose to report

the mean success rate rather than the median, as certain SMAC maps (especially Super Hard maps) commonly result in very success rates. Using the mean success rate better reflects the difficulty of these maps, as the median success rate can sometimes skew results to look more positive than they should. We set $\gamma = 0.99$ for all experiments.

Since our results are primarily obtained by applying recursive reasoning to FACMAC, we mostly kept the algorithmic implementation standards used in FACMAC for reproducibility. We use parameter sharing for all actor and critic networks to speed up learning. All actor, critic, mixer, and Q-networks have target networks.

**MAMuJoCo** The architecture of all deep Q-networks is an MLP with 2 hidden layers with 400 and 300 units respectively. In all actor-critic methods, the architecture of the shared actor and critic networks is an MLP with 2 hidden layers with 400 and 300 hidden units respectively. All hidden layers for all networks use ReLU activations. All critic networks provide raw outputs while actor networks have a tanh activation at the output. Actor networks and DQNs receive the local observations of that agent as an input, appended with a one-hot vector due to the parameter sharing. All centralized critics and mixing networks are conditioned on the global state provided by the environment.

Each episode has a maximum length of 1000 steps. The total training time for each algorithm is set to 2 million steps. To improve initial exploration, each agent takes 10,000 random steps at the beginning of each run. During training we apply uncorrelated, mean-zero noise with a standard deviation of 0.1 to further encourage exploration. Each agent has a replay buffer with a maximum size of 1 million and trains with a batch size of 100 after every new sample. Target networks are updated using Polyak averaging with $\tau = 0.001$. All neural networks are trained using the Adam optimizer (Kingma, 2014) with a learning rate of 0.001.

**SMAC** The architecture of all shared deep Q-networks is a DRQN with a recurrent layer comprised of a GRU with a 64-dimensional hidden state, with fully connected layers on either side. In all actor-critic methods, the architecture of all shared actors is a recurrent MLP comprised of a GRU with a 64-dimensional hidden state, with fully connected layers on either side. We train the GRU networks on batches of 32 fully unrolled episodes (with 0-padding to account for temporal mismatch between episodes). The architecture of all shared critic networks is an MLP with 2 hidden layers with 64 units. All networks use ReLU activations for the hidden layers. All actor critic methods select discrete actions using the Gumbel-Softmax estimator (De Boer et al., 2005) in order to turn continuous softmaxed logits into discrete one-hot actions while retaining the ability to backpropagate through the network. Actor networks and DRQNs receive the local observations of that agent as an input, appended with the last action taken by the agent, as well as a one-hot vector due to the parameter sharing. All agents use $\epsilon$-greedy action selection and we anneal $\epsilon$ from 0.5 to 0.05 over 50k training steps. The replay buffer contains the most recent 5000 episodes. All target networks are updated hard every 200 training steps. All networks are trained using Adam with a learning rate of 0.0025 for the actor network and 0.0005 for the critic network (except for QMIX which uses the learning rates specified in Samvelyan et al. (2019) as they have already been tuned for SMAC).

