# OpenReview forum: "Recursive Reasoning for Sample-Efficient Multi-Agent Reinforcement Learning"
_ICLR.cc/2026/Conference — ICLR 2026 Conference Withdrawn Submission_

### Official Review · Reviewer_f9Fp · 2025-10-26

**Soundness:** 2
**Presentation:** 2
**Contribution:** 2
**Rating:** 4
**Confidence:** 4

**Summary:**

This work tackles general-sum MARL by recursive reasoning, which corrects the miscoordination problem for previous methods using simultaneous policy updates, such as MAPPO and FACMAC. Theoretically, it converges to a local NE under some assumptions on the gradients. The experimental results are provided for cooperative settings and 1-level recursive reasoning in MA-MuJoCo and SMAC.

**Strengths:**

As far as I am aware, the theoretical convergence properties of MAPPO and MADDPG are not well known, despite having some empirical success in cooperative MARL. From that perspective, this work potentially provides insight into the NE convergence of simultaneous update approaches such as MAPPO. This could potentially be a valuable contribution as these approaches are more desirable in practice compared to techniques like HAPPO [1] which are slower in practice due to the sequential update scheme.

With that being said, these points are definitely not clear in the text, as it is positioned as “recursive reasoning” even though the agents are not doing any opponent modelling (see Weaknesses below).

Finally, there is some positive performance gap between ReMAPPO and MAPPO especially in SMACv2
for 20 agent tasks.

[1] Heterogeneous-Agent Reinforcement Learning (Zhong et, al. JMLR 2024)

**Weaknesses:**

* Overall, the motivation and approach has some gaps. The motivation for this work is recursive reasoning or theory of mind, but the problem setting is under CTDE. Thus, there is no opponent modelling where the policy considers the other agents’ actions or policies. The method the authors propose is more of a correction mechanism for simultaneous gradient updates.
* Related to the first point, and briefly mentioned in Strengths, it would require a significant re-write of the storyline if the authors want to focus on a more unbiased gradient updates for simultaneous update approaches, and not obfuscate their contributions with a slightly misleading interpretation of “recursive reasoning”.
* This work attempts to tackle general-sum games, but the experiments are limited to fully cooperative settings.
* The experimental results are limited to one level of recursive reasoning.
* Some strong baselines are missing such as HAPPO and HASAC, although I also acknowledge that improvement over MAPPO and FACMAC are more important for this work.
* There is no comparison to similar previous work such as PR2 [1] and GR2 [2].
* The writing structure is unorganized, as theoretical results should precede the experimental results.
* I think that the assumption that the other agents are stationary is only part of fictitious play, since it also assumes that the other agents are fixed at the empirical frequency of actions.

[1] Probabilistic Recursive Reasoning for Multi-Agent Reinforcement Learning (Wen et, al. ICLR 2019)
[2] Modelling Bounded Rationality in Multi-Agent Interactions by Generalized Recursive Reasoning (Wen et, al. IJCAI

**Questions:**

1. What is the purpose of recursive reasoning under CTDE?
1. What is the difference between Eq. 1-2 and iterative best response?
1. How does computation time scale with $k$?
1. Please address the points raised for Weaknesses.

---

### Official Review · Reviewer_R9qs · 2025-11-01

**Soundness:** 2
**Presentation:** 3
**Contribution:** 1
**Rating:** 2
**Confidence:** 4

**Summary:**

The paper proposes recursive reasoning for multi-agent policy optimization, where each agent updates by explicitly conditioning its gradient on teammates’/opponents’ just-updated policies within the same iteration. Concretely, it augments PPO/DPG objectives with a joint importance ratio, yielding practical algorithms (ReMAPPO, ReFACMAC, ReMADDPG) and a $k$-step recursion analysis with local convergence guarantees. Empirically on SMAX, SMAC, and MAMuJoCo, the method improves sample efficiency and final performance over strong baselines, particularly on harder cooperative tasks.

**Strengths:**

Thorough and rigorous theoretical analysis: the paper formalizes recursive reasoning with clear assumptions, derives generalized performance-difference bounds and the joint ratio, and analyzes $k$-step recursion with local convergence guarantees toward $\epsilon$-Nash under step-size and smoothness conditions. The proofs are carefully organized and provide actionable insight into how conditioning on $\pi'_{-i}$ reshapes gradients and stability, offering a solid conceptual foundation for the methodology.

**Weaknesses:**

- Limited comparisons to recent methods: the evaluation focuses on older baselines (MAPPO, FACMAC) and omits head-to-head results and differential analyses against contemporary agent-by-agent/coordinated PPO variants (CoPPO, HAPPO/HATRPO, A2PO), making it hard to assess the incremental value of the joint ratio $r'_i$ [1,2,3].
- Overlap with sequential agent-by-agent updates is under-analyzed: ReMAPPO’s “condition on $\pi'_{-i}$ within an iteration” resembles per-epoch agent-by-agent updating repeated across epochs (see the A2PO appendix), but the paper offers neither a clear theoretical separation nor empirical ablations to isolate order effects vs. recursion [3].

References

[1] Wu et al., “Coordinated Proximal Policy Optimization (CoPPO).”

[2] Kuba et al., “Trust Region Policy Optimisation in Multi-Agent Reinforcement Learning (HAPPO/HATRPO).”

[3] Wang et al., “Order Matters: Agent-by-agent Policy Optimization (A2PO).”

**Questions:**

- Can you provide head-to-head results and differential analyses against recent agent-by-agent/coordinated PPO variants (CoPPO, HAPPO/HATRPO, A2PO) under matched compute, seeds, and evaluation protocols?
- How is recursive conditioning on $\pi'_{-i}$ fundamentally different from agent-by-agent sequential updates within an epoch; under what conditions are they equivalent or provably distinct?
- How sensitive is training to the design of the joint ratio $r'_i$ and clipping—e.g., separate clipping for self vs. others, trust-region constraints, or adaptive step sizes?
- Could you report wall-clock time, FLOPs, and memory overhead for $k$-step recursion, and compare under equal-time vs. equal-sample budgets?

---

### Official Review · Reviewer_YmLP · 2025-11-02

**Soundness:** 2
**Presentation:** 2
**Contribution:** 2
**Rating:** 2
**Confidence:** 4

**Summary:**

The paper introduces ReMARL algorithms based on recursive reasoning on top of existing Multi-agent Reinforcement Learning baselines, specifically ReMAPPO, ReFACMAC, ReMADDPG that instead of updating each agents’ policy one time, it does that recursively in a way such that each agent responds to the update in other agents’ policy which as claimed in the paper results in more coordinated policies.

Paper contributions:
- They show how to practically incorporate recursive reasoning into existing MARL algorithms like MAPPO and FACMAC.
- Theoretically prove monotonic convergence of the methods to a local Nash equilibrium
- Test the method against existing baselines and show it achieves better performance in benchmarks like StarCraft 2 and MuJoCo.

**Strengths:**

- The experiments span multiple benchmark families (StarCraft II and Multi-Agent MuJoCo), and the results generally show consistent improvements.

- writing clarity: the theoretical exposition is clearer, the recursive updates are better formalized (Eqs. 1–2), and the higher-level convergence discussion (Section 6.1) is more polished.

**Weaknesses:**

I have previously reviewed a version of the paper, and although I appreciate some additions by the authors to improve the paper, some concerns still haven't been addressed in this version.

- The theoretical contributions largely build on previous work, particularly [1], by extending it from the two-player case to the general N-player setting. This extension appears relatively straightforward and raises questions about the novelty of the theoretical insight; see the extended discussion below.
- While the method shows clear improvements at recursion depth k=2, the marginal gains from deeper recursion levels are limited, especially when weighed against the significant increase in computational cost. This calls into question the practical utility of deeper recursion.
- The method is evaluated only in cooperative MARL settings. It remains unclear whether it can generalize effectively to competitive or mixed settings.
- The proposed method incurs a significantly higher computational cost (as discussed later), and for a fair comparison, results should be plotted using the number of gradient queries on the x-axis.
- The paper does not include comparisons with well-established methods such as Extragradient, nor is it mentioned in the related work section. This omission is particularly notable given that the foundational work the authors build upon [1] includes both comparisons and discussion of such methods.



Theory:

Previously, the paper stated that they extend what was done in [1] to the N player general sum game. Therefore, theoretical analysis in the paper extends what has been done in [1] from the 2-players setup to the n-player setting. Compare for example Theorem 6.2 in this paper to theorem 4.1 in [1]. This raises the question of how novel this theoretical result is and most importantly, in the current version they didn’t even mention the relation of the theoretical framework to [1] and they only reference it in one place as one of the recursive reasoning algorithms and used for GANs.

Furthermore, the analysis would benefit from a stronger connection to operator theory, which is directly relevant. For instance, the relationship between the largest eigenvalue and operator contractiveness is a key concept that is not sufficiently explored in the current presentation.

I also have concerns regarding the method’s robustness compared to well-established alternatives such as one-level or multi-level Extragradient methods, and related approaches in the Variational Inequality literature. Consider the simple min-max game min_x max_y initialized at (1,1). If one follows the steps described in Figure 1, the variables x and y may reach the solution (0,0), but the method does not remain there—it continues to oscillate. While proper step size tuning might mitigate this in such a simple example, the behavior reveals an underlying sensitivity to the problem structure and hyperparameter settings. This could pose challenges in more complex or ill-conditioned scenarios.


Implementation and results:

The paper does a good job in moving the base into more practical side on how to implement recursive reasoning into MAPPO, FACMAC and MADDPG and also testing the methods in complex MARL challenges such as SMAC and MAMuJoCo and shows performance improvements across all tested benchmarks.

However, the authors discuss that the method is computationally very expensive when k is increased, which is intuitive due to the additional gradient computations needed in the k-levels for each of the n-agents. In most shown cases, this increase is also not justified in significant  performance increase of higher levels compared to k=2. All of this questions the method's ability to scale in real applications and also why it is useful to carry on with all those levels in the first place.

Another lacking aspect is that the paper doesn’t compare to methods like lookahead[2] or extragradient[3] which require less gradient updates than the ReMARL method. The importance of comparison arises from that those methods also address the joint parameters dynamics and help to contract agents' policies in the shared space. The work done in [1] actually compares to both these methods and shows the relevance of the k iteration method to Extragradient. It is needed to show how the method performs against them, in a way where computational costs of each are highlighted.




----

[1] Zichu Liu and Lacra Pavel. Recursive reasoning in minimax games: A level k gradient play method. Advances in Neural Information Processing Systems, 35:16903–16917, 2022.

[2] M. Zhang, J. Lucas, J. Ba, and G. E. Hinton. Lookahead optimizer: k steps forward, 1 step back. In NeurIPS, 2019b.

[3] Galina M Korpelevich. The extragradient method for finding saddle points and other problems. Matecon, 12:747–756, 1976.

**Questions:**

Please see above.

---

### Official Review · Reviewer_PZBo · 2025-11-04

**Soundness:** 3
**Presentation:** 3
**Contribution:** 2
**Rating:** 6
**Confidence:** 2

**Summary:**

This paper introduces a simple but useful modification to MAPG methods: after taking the usual “naive” update, each agent immediately recomputes its own update while treating the teammates’ just‑updated policies as fixed. The authors instantiate this idea in both on‑policy (ReMAPPO) and off‑policy (ReFACMAC, ReMADDPG) settings, and back it with a convergence analysis that formalizes how deeper levels of recursion move you toward a local Nash equilibrium. Empirically, the method is competitive to strong baselines across SMAX, SMAC, and MAMuJoCo, typically learning faster and reaching higher final performance (e.g., ReMAPPO beats MAPPO on most SMAX maps; ReFACMAC is strong on hard SMAC maps and difficult MAMuJoCo tasks).

**Strengths:**

The paper is clearly written. The formulation in Eqs. (1)–(2) is simple and broadly applicable to both stochastic and deterministic policy‑gradient methods; the on‑policy instantiation (Eq. (4)) is particularly easy to implement on top of existing PPO/MAPPO pipelines.  The Generalized Semi‑Proximal Point Method view (Eq. (12)) plus Theorems 6.2–6.4 make the “anticipation of others’ updates” idea precise, very clean.

**Weaknesses:**

My main concern is an objective mismatch. The on‑policy surrogate keeps the advantage estimated under the current joint policy but reweights samples using the teammates’ updated action distributions; that means the quantity being optimized no longer faithfully estimates the true return against those updated policies. Appendix C also notes a state‑distribution mismatch that is ignored in practice. On the off‑policy side, the actors query critics at joint actions that can be far from the data support (after swapping in teammates’ updated actions), yet the paper claims no extra bias relative to non‑recursive baselines—this needs qualification or measurement. Concretely, I recommend re‑estimating advantages under the “perspective” joint policy (or using doubly‑robust / V‑trace style corrections or KL‑regularized objectives), and for off‑policy methods, quantifying critic extrapolation error and testing gradient‑direction alignment against Monte‑Carlo estimates of the true objective.

Coverage of related work is also too narrow. Beyond LOLA/POLA and M‑FOS, please situate the method alongside K‑Level Policy Gradients (KPG), Recursive Reasoning Graph (R2G), Generalized Recursive Reasoning (GR2), Policy‑Space Response Oracles (PSRO), and Stable Opponent Shaping (SOS), clarifying similarities and differences in goals, computation, and stability. It would also help to spell out the training‑time assumption that you can evaluate other agents’ updated action probabilities, and discuss robustness when those distributions are only approximated; the conclusion briefly flags this but does not analyze it. Finally, add fairness checks (cross‑setting comparisons) and report the compute overhead of deeper recursion, since the ablation suggests most of the gain is already achieved by two levels. With these clarifications—and especially by fixing the objective mismatch—the paper would be stronger.

4.2 The “no additional bias” statement would benefit from a lemma with approximation terms (e.g., Bellman residuals under action substitution) or an empirical conservative Q ablation.

**Questions:**

For ReFACMAC/ReMADDPG, how far (in action space) do teammates’ updates move between recursions, and how does Q‑error grow with that distance? A plot of error vs. \mu’{-i}(s)-\mu{-i}(s) would be something to consider?

---

### Official Review · Reviewer_kRZz · 2025-11-06

**Soundness:** 3
**Presentation:** 2
**Contribution:** 2
**Rating:** 4
**Confidence:** 4

**Summary:**

The paper proposes a recursive reasoning framework for multi-agent policy gradient methods. Each agent refines its update by conditioning on the updated policies of the other agents, yielding ReMAPPO, ReFACMAC, and ReMADDPG. The authors extend the performance-difference argument, modify importance ratios for joint updates, and prove that finite-level recursion converges monotonically toward a local Nash equilibrium under Lipschitz conditions assumptions.

**Strengths:**

1. The core recursive update $\theta_i^{\mathrm{Re}} \leftarrow \theta_i+\eta_i \nabla_{\theta_i} J_i\left(\theta_i, \theta_{-i}^{\prime}\right)$ is a minimal modification of standard MARL gradients yet explicitly breaks the fictitious-play assumption, giving a principled way to anticipate co-learners' updates inside the same step.

2. The paper re-derives the MAPPO surrogate to include a joint importance ratio $r_i^{\prime}(s, a)=\frac{\pi_i^{\prime}\left(a_i \mid s\right)}{\pi_i\left(a_i \mid s\right)} \cdot \frac{\pi_{-i}^{\prime}\left(a_{-i} \mid s\right)}{\pi_{-i}\left(a_{-i} \mid s\right)}$, which is the object when everyone updates; this makes the recursion consistent with trust-region/clipping objectives.

3. The higher-level analysis shows that if gradients are Lipschitz in opponents' parameters and step sizes satisfy $\eta<\frac{1}{L(N-1)}$, then finite recursive depth yields a sequence that contracts toward an $\varepsilon$-Nash equilibrium, so the method is not just heuristic but tied to a clear stability condition.

**Weaknesses:**

1. The framework's core mechanism relies on each agent having access to the updated policy parameters (e.g., $\theta_{-i}^{\prime}$ or $\pi_{-i}^{\prime}$ ) of **all other agents** during the training step. This is a significantly stronger assumption than standard Centralized Training, Decentralized Execution (CTDE) frameworks, which may only require a centralized critic. This reliance poses a major barrier to practical deployment in systems with communication latency or privacy constraints.

2. The paper's gains in "sample efficiency" are achieved at the direct expense of "computational efficiency". A $k$-level recursive update requires $k$ separate gradient computations within a single update step. For the primary methods where $k=2$, this at least doubles the computational cost per update. The paper does not provide any wall-clock time comparisons to justify this trade-off.

3. The theoretical convergence proofs (e.g., Theorem 6.2, 6.3) rest on an idealized, noise-free assumption. The analysis does not account for how the convergence guarantees would be affected by real-world factors such as estimation error, communication noise, or latency when accessing the other agents' updated policies $(\theta_{-i}^{(k-1)})$.

**Questions:**

1. From my perspective of view, the ReMAPPO loss function (Equation 4) introduces a potential source of bias that is not quantified. It mixes an advantage function $A_i^\pi(s, a)$-which is estimated using data from the old joint policy $\pi$-with a **new** importance sampling ratio $r_i^{\prime}(s, a)$ that is based on the post-update joint policy $(\pi_i^{\prime}, \pi_{-i}^{\prime})$. So I think ReMAPPO introduces a more complex, non-quantified mismatch on top of the original mismatch in PPO. Can you explain this for me?

2. Regarding the off-policy variants, the paper asserts that the recursive update introduces "no additional bias" over its non-recursive counterparts . Can this strong claim be formally proven? How is this possible when the recursive step itself (Equation 7) relies on a critic (the $Q_i$ function) that is already known to be biased from offpolicy data, target bootstrapping, and the non-stationarity of other agents?

3. The paper provides a comprehensive theoretical framework for higher-order reasoning ($k>2$). However, given the empirical results in Section 6.3 which show diminishing, and in some cases negative, returns (with most benefits materializing at $k=2$), what is the practical justification for this complex higher-order framework?

4. How does the framework's performance scale with a large number of agents $(N)$ ? Specifically, could the product of $N-1$ policy ratios in the ReMAPPO loss lead to numerical instability (i.e., vanishing or exploding gradients)? Furthermore, does the centralized critic in ReFACMAC, which takes the joint actions of all agents as input, suffer from input-dimension scalability issues as $N$ increases?

---

### Note · Authors · 2025-11-24

**Comment:**

We sincerely thank the reviewers and AC for their valuable time in considering our work. We have chosen to withdraw our submission from ICLR 2026 and we will integrate your helpful insights into a future submission.

**Withdrawal Confirmation:**

I have read and agree with the venue's withdrawal policy on behalf of myself and my co-authors.